# A taxon-restricted duplicate of *Iroquois3* is required for patterning the spider waist

**Emily V. W. Setton** [1]*, **Jesús A. Ballesteros**[2], **Pola O. Blaszczyk**[1], **Benjamin C. Klementz**[1], **Prashant P. Sharma**[1]*

**1** Department of Integrative Biology, University of Wisconsin-Madison, Madison, Wisconsin, United States of America, **2** Department of Biology, Kean University, Union, New Jersey, United States of America

* setton@wisc.edu (EVWS); prashant.sharma@wisc.edu (PPS)

**Data Availability Statement:** All image data are all contained within the paper and/or Supporting Information files. Outputs of analyses and all reagents used are available in the supporting

## Abstract

The chelicerate body plan is distinguished from other arthropod groups by its division of segments into 2 tagmata: the anterior prosoma ("cephalothorax") and the posterior opisthosoma ("abdomen"). Little is understood about the genetic mechanisms that establish the prosomal-opisthosomal (PO) boundary. To discover these mechanisms, we created high-quality genomic resources for the large-bodied spider *Aphonopelma hentzi*. We sequenced specific territories along the antero-posterior axis of developing embryos and applied differential gene expression analyses to identify putative regulators of regional identity. After bioinformatic screening for candidate genes that were consistently highly expressed in only 1 tagma (either the prosoma or the opisthosoma), we validated the function of highly ranked candidates in the tractable spider model *Parasteatoda tepidariorum*. Here, we show that an arthropod homolog of the Iroquois complex of homeobox genes is required for proper formation of the boundary between arachnid tagmata. The function of this homolog had not been previously characterized, because it was lost in the common ancestor of Pancrustacea, precluding its investigation in well-studied insect model organisms. Knockdown of the spider copy of this gene, which we designate as *waist-less*, in *P. tepidariorum* resulted in embryos with defects in the PO boundary, incurring discontinuous spider germ bands. We show that *waist-less* is required for proper specification of the segments that span the prosoma-opisthosoma boundary, which in adult spiders corresponds to the narrowed pedicel. Our results demonstrate the requirement of an ancient, taxon-restricted paralog for the establishment of the tagmatic boundary that defines Chelicerata.

## Introduction

Functional understanding of the evolution of animal body plans is frequently constrained by 2 bottlenecks. First, developmental genetic datasets and functional toolkits are often asymmetrically weighted in favor of lineages that harbor model organisms, to the detriment of phylogenetically significant non-model groups. Second, models of ontogenetic processes that are grounded in model systems vary in their explanatory power across diverse taxa, both as a function of phylogenetic distance, as well as the evolutionary lability of different gene regulatory

information files. Raw sequence data for the tarantula transcriptomes used in this work are available via NCBI SRA (under PRJNA1105064). All other resources used have been previously published (i.e. genomes, transcriptomes, protocols).

**Funding:** This work was supported by the National Science Foundation (IOS-1552610 and IOS-2016141 to PPS) (nsf.gov). Additional support to EVWS came from The National Science Foundation Graduate Research Fellowship (DGE-1747503 to EVWS) (nsf.gov). The funders had no role in study design, data collection and analysis, decision to publish, or preparation of the manuscript.

**Competing interests:** The authors have declared that no competing interests exist.

**Abbreviations:** AP, antero-posterior; DGE, differential gene expression; dsRNA, double-stranded RNA; GRN, gene regulatory network; HCR, hybridization chain reaction; PFA, paraformaldehyde; PO, prosomal-opisthosomal; RNAi, RNA interference; WGD, whole genome duplication.

networks (GRNs) [1–4]. In Arthropoda, understanding of morphogenesis, as well as the evolutionary dynamics of underlying GRNs, is largely grounded in hexapod models and, particularly, holometabolous insects. Candidate gene approaches derived from studies of insect developmental genetics have thus played an outsized role in understanding the mechanisms of arthropod evolution, with emphasis on processes like segmentation, limb axis patterning, and neurogenesis [5–10]. However, the candidate gene framework has its limits in investigations of taxon-specific structures (e.g., spider spinnerets, sea spider ovigers) [11–13], or when homologous genes or processes do not occur in non-model taxa (e.g., *bicoid* in head segmentation) [7,14].

These limits are accentuated in Chelicerata (e.g., spiders, scorpions, mites, horseshoe crabs), the sister group to the remaining arthropods. The bauplan of most chelicerates consists of 2 tagmata, the anterior prosoma (which bears the eyes, mouthparts, and walking legs) and the posterior opisthosoma (the analog of the insect abdomen). Even at this basic level of body plan organization, differences in architectures are markedly evident between chelicerates and the better-studied hexapods. The chelicerate prosoma typically has 7 segments and includes all mouthparts and walking legs, whereas the insect head has 6 segments and bears only the sensory (antenna) and gnathal appendages (mandible, maxilla, labium); locomotory appendages of insects occur on a separate tagma, the thorax [15].

Comparatively little is known about how these functional groups of segments are established in chelicerates, by comparison to their insect counterparts. Due to the phylogenetic distance between hexapods and chelicerates, homologs of insect candidate genes that play a role in tagmosis can exhibit dissimilar expression patterns or incomparable phenotypic spectra in gene silencing experiments in spiders, a group that includes the leading models for study of chelicerate development [13,14,16–18]. A further complication is the incidence of waves of whole genome duplications (WGDs) in certain subsets of chelicerate orders, such as Arachnopulmonata, a group of 6 chelicerate orders that includes spiders [19–22]. The retention of numerous paralogous copies that diverged prior to the Silurian represents fertile ground for understanding evolution after gene duplication but also presents the potential barrier of functional redundancy or replacement between gene copies. Accordingly, there are few functional datasets supporting a role for lineage-specific gene duplicates in the patterning of arachnid body plans [23,24].

To advance the understanding of chelicerate body plan patterning and address possible roles for retained paralogs in chelicerate tagmosis, we generated transcriptional profiles of prosomal and opisthosomal tissues of a large-bodied spider (a tarantula), across developmental stages pertinent to posterior patterning. We applied differential gene expression (DGE) analyses to identify taxon-specific gene duplicates that were differentially expressed across the prosomal-opisthosomal (PO) boundary and screened candidates using an RNA interference (RNAi) gene silencing approach. Through this approach, we were able to identify one of the 5 spider homologs of Iroquois (*Iroquois4 sensu* [25]; *Iroquois3-2, sensu* [26]) as playing a role in patterning the segments spanning the PO boundary. Our results provide a functional link between an unexplored gene copy restricted to non-pancrustacean arthropods and the boundary between the tagmata of chelicerates.

## Results

### Differential gene expression, RNAi screen, and identification of *waist-less*

To understand the genetic basis of posterior patterning in spiders, we aimed to generate tissue-specific transcriptomes of spider embryos. The leading model system for spider development, *Parasteatoda tepidariorum*, proved challenging in this regard, due to the small size of its

embryos (500 μm) and the high internal pressure of the egg. We therefore generated DGE datasets for the tarantula *Aphonopelma hentzi*, which features large and synchronous broods, and embryos with large diameter (2.4 mm) and low internal pressure [27]. We dissected clutches of synchronously developing tarantula embryos and generated RNA-seq libraries for the labrum, chelicera, pedipalp, walking leg, book lung, anterior spinneret, and posterior spinneret. Dissected tissues comprised the whole appendage and attached section of body wall. This protocol was performed for 3 developmental stages, encompassing establishment and differentiation of posterior appendages (e.g., book lungs and spinnerets) [27]. DGE analysis identified 5,429 to 14,094 genes (stage 9: 7,609; stage 10: 5,429; stage 11: 14,094) as consistently differentially expressed across segments in an all-versus-all comparison ($p \leq 0.05$; LFC $\geq 1$) (Fig 1A). To identify genes that may play an important role in posterior patterning, we assessed the top 100 most differentially expressed genes in all-by-all comparisons for each developmental stage and screened candidates that were consistently highly expressed in only the prosomal or opisthosomal segments in at least 2 stages (stage 9: 67; stage 10: 53; stage 11: 92) (S1 Fig). We prioritized 12 genes for functional screening based on their expression profiles (S1 Table

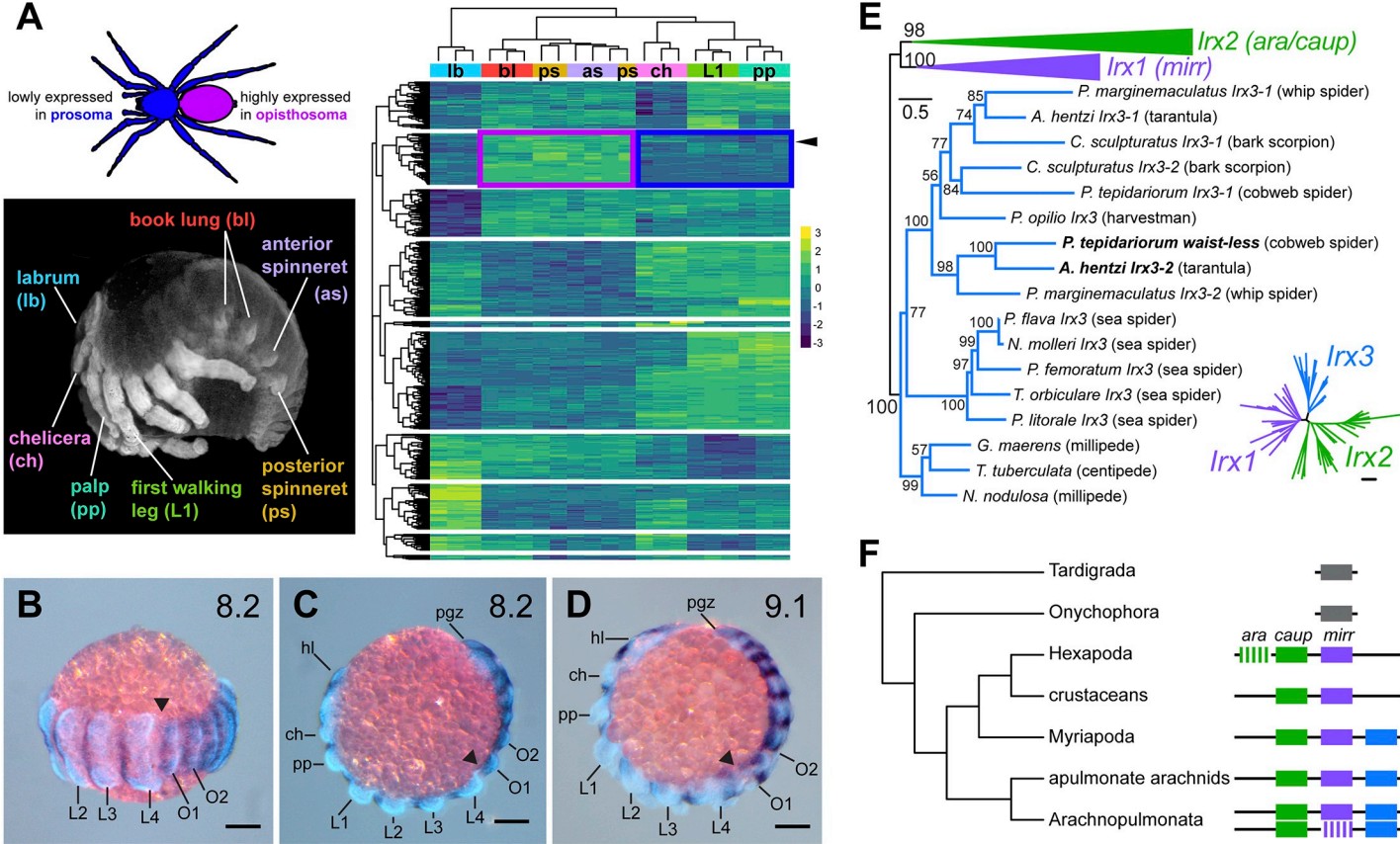

**Fig 1. Overview of RNA-seq design, candidate gene identification, and the ortholog identification within the *Iroquois* gene family.** (**A**) Tissues from regions representing major morphological characters along the antero-posterior (AP) axis were dissected from developing *Aphonopelma hentzi* embryos for mRNA sequencing. DGE analysis of RNA-seq libraries generated region-specific profiles to enable the identification of genes both lowly expressed in the prosoma (blue box) and highly expressed in the opisthosoma (purple box). Arrowhead indicates the ortholog of spider *waist-less*. Heatmap is based upon stage 10 embryos; all transcriptional profiles are provided in S1 Fig. (**B-D**) Expression of *waist-less* in limb bud stage embryos of *Parasteatoda tepidariorum*, counterstained for Hoechst. Note the higher expression level in the opisthosoma compared to the prosoma. (**E**) Maximum likelihood gene tree of *Iroquois2/3* homologs of Panarthropoda, rooted on Onychophora. Colored branches correspond to different orthologs, following F. Boldface text indicates spider *waist-less* orthologs. Inset: Full unrooted gene tree of Iroquois homologs. (**F**) Inferred evolutionary history of *Iroquois* gene duplications in Chelicerata. Scale bar: 100 μm. Complete dataset for heatmap in panel A is provided in S3 Data.

and S2 and S3 Figs). Candidate selection emphasized transcription factors (*n* = 8/12), with the remaining candidates comprising a secreted protein (*spaetzle*), a nucleotidyltransferase (*Mab21-1*), and 2 genes of unknown function (*Ahen-TRINITY_DN6222_c0_g1* and *Ahen-TRINITY_DN3695_c1_g1*).

Due to the lack of gene silencing tools in the tarantula, we performed functional screening of candidate genes in the house spider *Parasteatoda tepidariorum* (S2 Table), following established protocols [17,28–30]. Of the 12 candidates, 10 yielded no discernable phenotype, paralleling outcomes of recent RNAi screens in this system (S1 Table) [13]. Initial identification of phenotypes from RNAi screens was performed by visual assessment of morphology at developmental stages 8 to 13, following the established *P. tepidariorum* staging system [31], with embryos sampled from viable egg clutches 2 to 5 (until females stopped laying). The broad range of stages was chosen to encompass establishment of the PO boundary, addition of opisthosomal segments, and limb bud development in both tagmata. Within this developmental window, we looked for morphological defects in the limb buds, the segments spanning the PO boundary, and the opisthosomal tissues. A subset of each cocoon laid during the RNAi screen was raised to at least the first instar to assess the incidence of postembryonic defects. Genes highly expressed in the prosoma were included in the initial screen toward possible identification of transcripts responsible for repressing opisthosomal identity. However, none of the prosoma-biased genes resulted in a phenotype and were not further pursued (S1 Table and S2 Fig).

The 2 RNAi experiments resulting in phenotypes both targeted genes predicted to be enriched in the opisthosoma. One candidate that generated RNAi phenotypes in a small number of embryos was annotated as a GATA transcription factor. RNAi against the GATA homolog (*pannier2*) resulted in opisthosomal defects with low phenotypic penetrance, precluding analysis of large numbers of affected embryos (described below). The second candidate was annotated as a member of the Iroquois complex of homeobox genes (Fig 1B–1E). Previously identified as "*Iroquois4*" in a recent survey of homeobox family duplications [25], this transcription factor is not orthologous to the identically named vertebrate homolog Iroquois4 nor is its homology to its 2 insect homologs (*mirror* and *arucan/caupolican*) understood [32]. To forfend redundancy of nomenclature within the chelicerate Iroquois complex, we rename the differentially expressed spider copy (previously, "*Iroquois4*") *waist-less* (*wsls*), reflecting the phenotypic spectrum described below. Due to the higher penetrance of RNAi against *waist-less* and the ensuing ability to interrogate patterning of the PO boundary, this gene became the focal point of the study.

## Evolutionary history of panarthropod Iroquois homologs

To better understand the evolutionary history of this gene in arthropods, we inferred a gene tree of the Iroquois family, surveying genomes and developmental transcriptomes of 4 arachnopulmonates (arachnids that share a WGD; 2 spiders, a whip spider, and a scorpion), 6 non-arachnopulmonate chelicerates (chelicerates with an unduplicated genomes; 5 sea spiders and a harvestman), 4 myriapods (sister group to chelicerates with unduplicated genomes; 2 centipedes, 2 millipedes), 3 crustaceans, and 12 hexapods. The gene tree topology (Figs 1E and S4) recovered *Iroquois1*, *Iroquois2*, and *Iroquois3* homologs as 3 separate clusters, with maximal nodal support for *Iroquois3*. Whereas representatives of all major arthropod lineages bore *Iroquois1* and *Iroquois2* homologs, the cluster corresponding to *Iroquois3* was comprised only of myriapod and chelicerate taxa (Fig 1E).

To polarize the evolutionary history of the Iroquois complex, we examined the organization of Iroquois homologs in well-annotated genomes of Panarthropoda (Fig 1F and S3 Table).

Whereas a single Iroquois homolog occurs in high-quality genomes of Tardigrada and Onychophora, chromosomal-level genomes of Myriapoda and apulmonate Chelicerata exhibited 3 Iroquois homologs arranged contiguously on single scaffolds, consistent with an origin of the arthropod Iroquois genes via 2 tandem duplications. Chromosomal-level genomes of spiders recovered 5 to 6 Iroquois copies, with homologs of *Iroquois1*, *Iroquois2*, and *Iroquois3* occurring on 2 separate scaffolds, consistent with WGD in the arachnopulmonate common ancestor. The ancestral arrangement of the 3 Iroquois homologs was observed to be reordered in 1 cluster in the spider *Dysdera sylvatica* (see also [26]). In support of this result, non-arachnopulmonate chelicerates (e.g., the harvestman, sea spiders) bore 3 Iroquois homologs in the gene tree (1 homolog of *mirror*, 1 homolog of *araucan/caupolican*, and 1 homolog of *Iroquois3*), whereas spiders and scorpions bore up to 6 Iroquois homologs due to an arachnopulmonate-specific WGD. *P. tepidariorum* bore only 5 Iroquois homologs due to the loss of 1 *mirror* copy (S4 Fig).

The absence of *Iroquois3* in all sampled representatives of hexapods and crustaceans is consistent with a loss of this gene in the branch subtending Pancrustacea. Additionally, the duplication and subdivision of *Iroquois2* into *araucan* and *caupolican* is limited to a subset of flies (e.g., *Drosophila melanogaster*), not all Diptera (e.g., *Calliphora vicina*, *Anopheles gambiae*, *Culex pipiens quinquefasciatus*) (Fig 1F).

## Expression of *waist-less* in *Parasteatoda tepidariorum*

Expression of spider *waist-less* (formerly "*Irx4*", sensu [25]) was previously reported for selected stages of development, and a segmentation function had been suggested due to the segmentally reiterated stripes of expression [25,26,33]. We first surveyed *waist-less* expression across the embryogenesis of *P. tepidariorum*. The earliest expression was detected at stage 5 as a ring around the germ disc and accords with previously reported expression for stage 5 [34] (arrowhead in S5A Fig). Dynamic expression was recovered at stage 6 (dorsal field stage), corresponding to a stripe of expression in the outer margin of the embryo, in addition to a separate domain of expression in the presumptive growth zone (arrowheads in S5B and S5C Fig). Subsequent stages exhibited additional stripes generated at the posterior terminus, corresponding to presumptive segments of the prosoma (S5D–S5F Fig). In the transition from the dorsal field stage to the germ band stage (stage 7), expression of *waist-less* decreased in the growth zone and the strongest expression domains corresponded to the segmentally iterated stripes of the prosoma (S5G and S5H Fig). At stages 8 and 9, expression is notably stronger in opisthosomal segments compared to prosomal segments, due to the incidence of weaker domains bridging the segmentally iterated stripes of *waist-less* in the opisthosoma (Figs 1B–1D and S5I–S5N) and corroborating the stronger posterior expression predicted by DGE (S3 Fig). Additional expression domains include a "V" shape in the anterior head beginning at stage 8.2 (S5I–S5K Fig). At stage 9.2, expression appears in the lateral body wall, together with a distinct, distal point of expression in the prosomal appendages. At this stage, the "V" of expression in the anterior head becomes a pair of arcs, curved inward toward each other approaching the ventral midline and comprising the medial head region that lies anterior to the cheliceral limb buds (S5L–S5N Fig). At stage 10.1, the crescents of expression on each side of the developing head become more concentrated. The segmentally repeated stripes of expression are still maintained ventrally but with the stripes no longer of uniform strength across the germ band. At this stage, increased expression is seen in the opisthosomal appendages (S5O–S5Q Fig). Gene expression at stage 10.2 expression is similar to 10.1 but with increased localization to the lateral margins and appendage primordia of the opisthosoma (S5R–S5T Fig). Stage 11 embryos exhibit increased division of the stripes across the width of the germ band and continued strong expression in the lateral part of the opisthosoma (S5U–S5W Fig).

## Knockdown of *waist-less* disrupts the prosoma-opisthosoma boundary in a spider

To assess the function of *waist-less*, RNAi was performed using established protocols [17,28–30]. Parental RNAi against *Ptep-waist-less* via maternal injections of double-stranded RNA (dsRNA) resulted in a phenotypic spectrum affecting the PO boundary. Validation of knockdown was assessed using colorimetric in situ hybridization (S6 Fig). Phenotypes were scored in embryos stage 8.1 or later, when morphological landmarks are present, and designated into 2 classes. Class I phenotypes (17.2%; *n* = 41) exhibited a PO boundary defect, consisting of reduction of the first opisthosomal and the fourth walking leg segments (Fig 2E–2E'). In later stages, the embryo developed as a discontinuous germ band, with no embryonic tissue in the region corresponding to the posterior prosoma and the anterior opisthosoma (Fig 2F–2F'). Class II phenotypes (31.9%; *n* = 76) exhibited reduction of embryonic tissue spanning the anterior opisthosoma up to the middle of the prosoma (walking leg II) (Fig 2G–2G'). Class II phenotypes also exhibited a discontinuity at the boundary between tagmata (Fig 2H–2H'), but additional defects observed in the prosoma included fusion of adjacent limb buds, bifurcated pedipalps, and smaller chelicerae (Fig 2G–2G'). Few embryos exhibiting *Ptep-waist-less* phenotypes completed development; in later stages of embryogenesis, a small number of embryos were observed with discontinuous prosoma and opisthosoma (*n* = 3/55) (S7D and S7E Fig). In

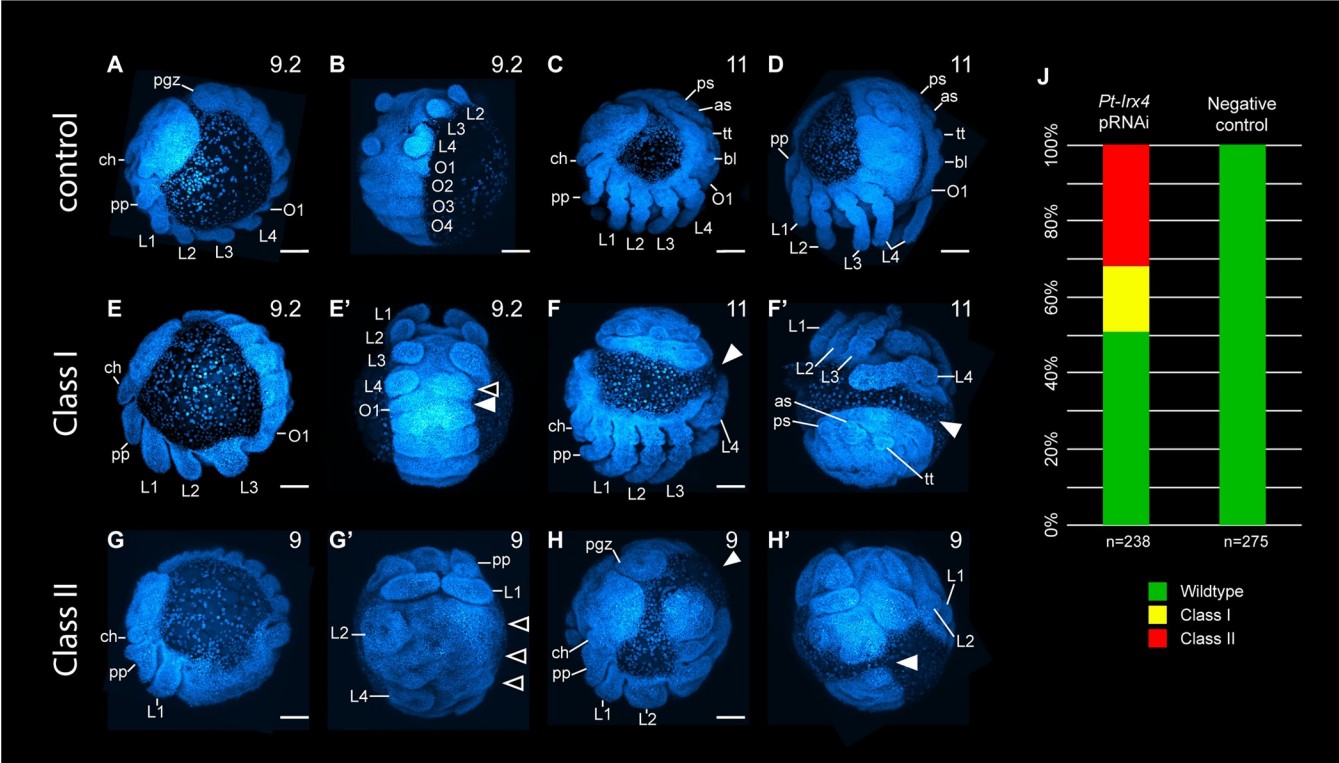

**Fig 2. Phenotypic spectrum of *Ptep-waist-less* maternal RNAi. (A-D)** Wild-type development of *P. tepidariorum* in negative control experiments. **(E-F')** Class I *Ptep-waist-less* RNAi embryos exhibit reduction or loss of L4 segment (**E'**, open arrowhead) or disruption of both L4 and anterior opisthosomal segments (**E'**, solid arrowhead). Some Class I embryos also exhibit discontinuous germ bands (**F-F'**, solid arrowhead). **(G-H')** Class II *Ptep-waist-less* RNAi embryos exhibit defects spanning the L2 or L3 segment to anterior opisthosomal segments, as well as bifurcating pedipalps and reduced chelicerae (**G, G'**). In the same manner as Class I phenotypes, some Class II phenotypes exhibit discontinuous germ bands (**H, H'**, solid arrowhead). **(J)** Phenotypic distribution of *Ptep-waist-less* RNAi and negative control embryos. Abbreviations: as, anterior spinneret; bl, book lung; ch, chelicera; L1-L4, walking legs 1–4; O1-O4, opisthosomal segments 1–4; pgz, posterior growth zone; pp, pedipalp; ps, posterior spinneret; tt, tubular trachea. Scale bars: 100 μm. The data underlying the graphs shown in the figure can be found in S1 Data.

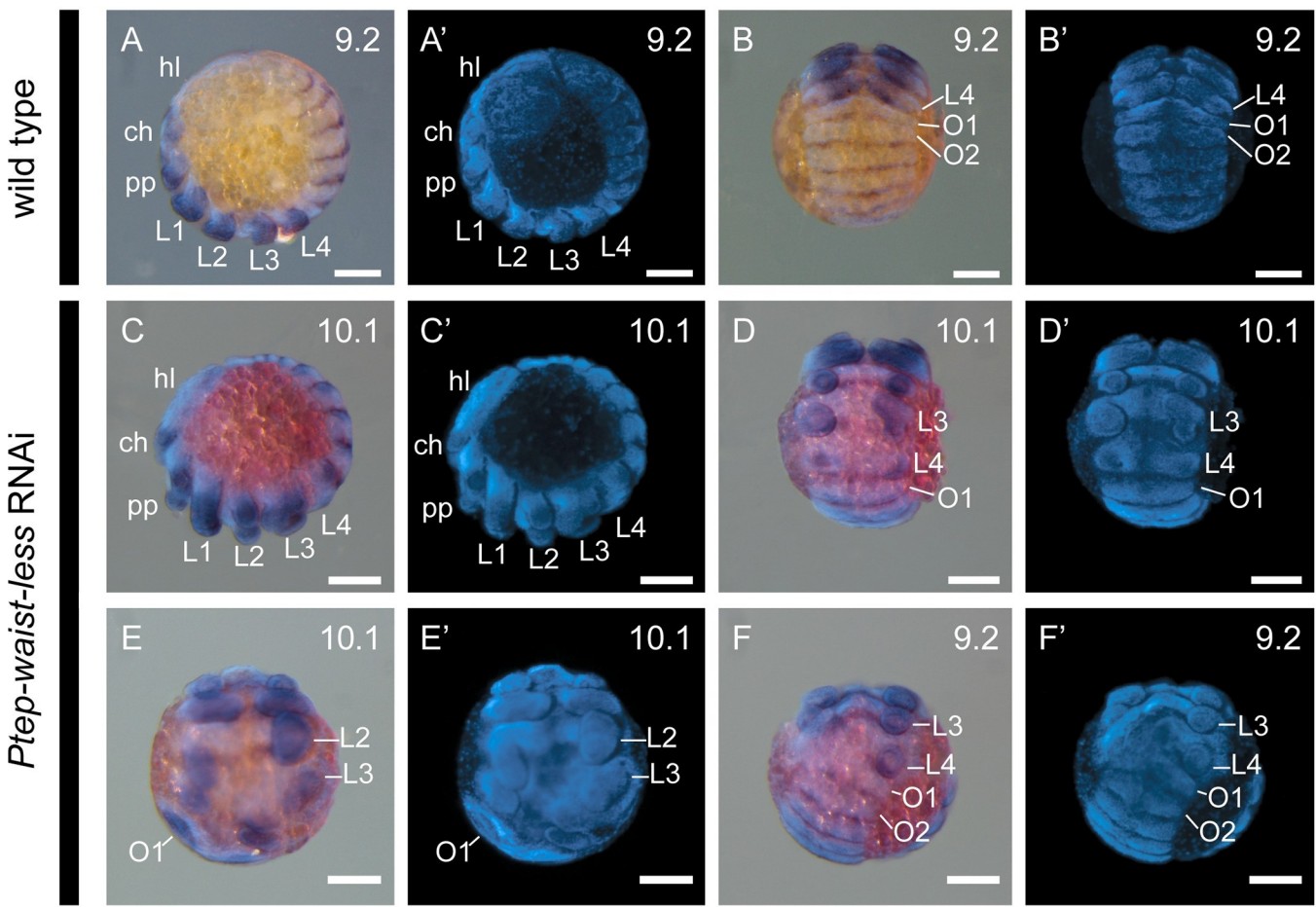

**Fig 3. Effects of *Ptep-waist-less* RNAi affect segments spanning the prosoma-opisthosoma boundary.** (**A-B'**) Wild-type embryos express the segmental marker *engrailed-1* (*en1*) in the posterior boundary of each segment; the limb-patterning gene *Distal-less* (*Dll*) is expressed in the distal part of each appendage. (**C-F'**) *Ptep-waist-less* RNAi embryos show disruption of segments at the prosoma-opisthosoma boundary (*en1* expression lost or disrupted in L2-O1) and loss or reduction of L2-L4 appendages (*Dll* missing or disrupted). (**A'-F'**) Hoechst counterstains of embryos in A-F. RNAi embryos have been overstained to ensure detection of riboprobes. Abbreviations: hl, head lobe. Other abbreviations as in Fig 2. Scale bars: 100 μm.

a separate RNAi trial, we observed 3 *Ptep-waist-less* RNAi postembryos, which exhibited the mildest phenotypic defect (truncated L4 segment on 1 side of the body; *n* = 3/59) (S7F Fig).

To assess the identity of the territories impacted by *Ptep-waist-less* RNAi, we assayed a segmental boundary marker (*engrailed-1*; *en1*) and a distal appendage marker (*Distal-less*; *Dll*) (Fig 3) [35,36]. Individually, *engrailed-1* is expressed as iterated stripes in the posterior compartment of each segment; *Distal-less* is expressed in the distal territory (trochanter to tarsus) of the appendage proximo-distal axis. In *Ptep-waist-less* RNAi embryos, expression of *Ptep-en1* was lost at the PO boundary (L4 walking leg segment and O1 opisthosomal segment in Class I embryos), with concurrent loss or diminution of *Ptep-Dll* expression (Fig 3C and 3D). Additional posterior walking leg primordia and their corresponding *engrailed-1* stripes were lost in Class II embryos (Fig 3E and 3F). Separately, we assayed *Ptep-en1* and the Hox gene *Sex combs reduced-1* (*Scr1*), which is most strongly expressed in the distal territories of the L3 and L4 limb buds (Fig 4) [21]. *Ptep-waist-less* RNAi embryos exhibited specific and consistent reduction in *Ptep-Scr1*, concomitant with disruption of *Ptep-en1* stripes in this territory (Fig 4C and 4D).

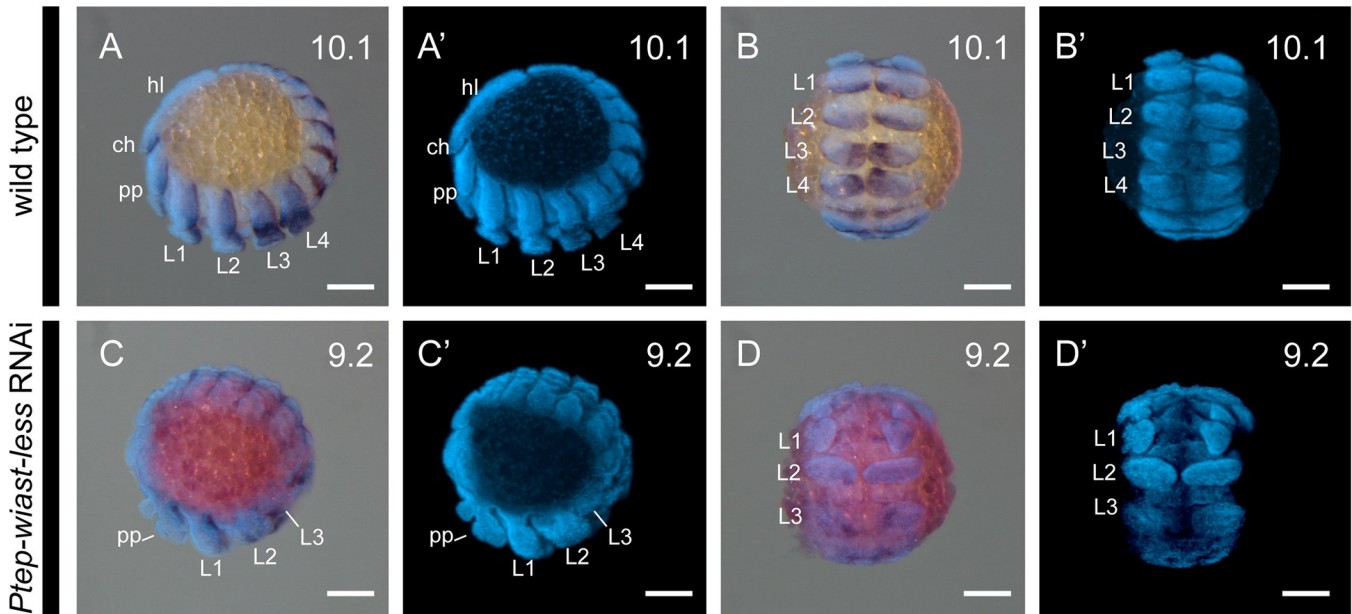

**Fig 4. RNAi against *Ptep-waist-less* affects the posterior prosomal segments and is not associated with homeosis. (A-B')** Wild-type embryos express *engrailed-1* (*en1*) at the posterior boundary of each segment. The Hox gene *Sex combs reduced-1* (*Scr1*) is strongly expressed in the distal territories of L3 and L4 limbs. **(C-D')** *Ptep-waist-less* phenotypes show disrupted segmentation (*en1* expression lost or disrupted) and concomitant loss of the third and fourth walking legs (Class II phenotype, partial loss of L3 and L3 segmental boundary). **(A'-D')** Hoechst counterstains of embryos in A-D. RNAi embryos have been overstained to ensure detection of riboprobes. Abbreviations as in Figs 2 and 3. Scale bars: 100 μm.

These results support the interpretation that the most pronounced effects of *Ptep-waist-less* RNAi target the segments spanning the PO boundary.

To understand the extent of PO boundary defects in RNAi phenotypes, we surveyed embryos for the expression of the ventral midline marker *short gastrulation* (*sog*), whose expression has been well characterized in *P. tepidariorum* [28,37], as well as the opisthosomal dorsal marker *pannier2* (recovered in our DGE analysis and discussed below). In the opisthosoma, *Ptep-waist-less* is expressed in a field of cells overlapping the *Ptep-pnr2*-positive territory in the dorsal margin, as well as more lateral cells (in addition to the stripes of expression in the ventral ectoderm described previously) (S8A'–S8C' Fig). There is no overlap between the expression of *Ptep-sog* and the expression of either *Ptep-pnr2* or *Ptep-waist-less*, except for the ventral stripes of *Ptep-waist-less* expression (Figs 5A–5A''' and S8A''–S8C''). In *Ptep-waist-less* Class I RNAi embryos with discontinuous germ bands, the ventral midline expression of *Ptep-sog* was rendered discontinuous at the prosoma-opisthosoma boundary (Fig 5B' and 5B'''). In the same RNAi embryos, *Ptep-pnr2* was ectopically expressed at the narrowest point of the constricted germ band, in the territory that corresponded to the deleted ventral midline (Fig 5B'' and 5B'''). Class I embryos of *Ptep-waist-less* RNAi without discontinuity of the antero-posterior (AP) axis (interpreted to mean a mild loss-of-function phenotype) retained *Ptep-pnr2* expression in the opisthosoma, but the expression domain of *Ptep-pnr2* was rendered irregular (S9 Fig).

Disruption of *sog* at the PO boundary in the *Ptep-waist-less* RNAi phenotype could alternatively reflect (a) a gap segmentation function localized to the boundary of the tagmata, or (b) a localized defect in proper dorso-ventral patterning [28]. To test whether the interruption of *sog* is consistent with gap segmentation phenotypes, we repeated RNAi experiments against the previously identified spider gap gene *Ptep-Sp6-9* [30] and assayed the resulting phenotypes

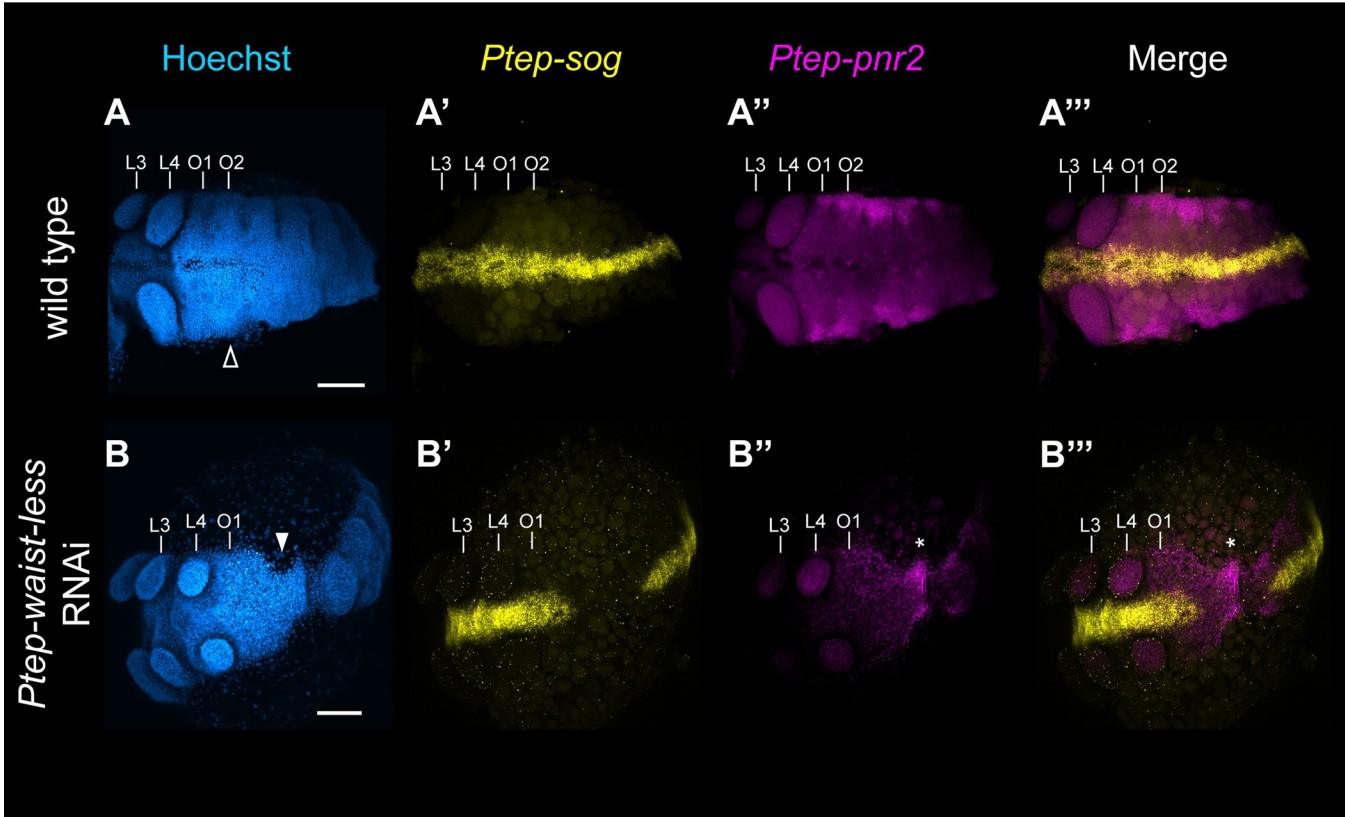

**Fig 5. Knockdown of *Ptep-waist-less* incurs a gap phenotype.** (A-A''') Wild-type stage 9 *P. tepidariorum* embryo with a continuous germ band (**A**, open arrowhead); continuous expression of *Ptep-sog* along the ventral midline of the antero-posterior axis (**A'**); and expression of *Ptep-pnr2* in the lateral margins of the opisthosoma (*n* = 10/10) (**A"**). (**B-B'''**) *Ptep-waist-less* RNAi stage 9 embryos exhibit interrupted expression of the ventral marker *Ptep-sog* (**B'**) in regions affected by *Ptep-waist-less* knockdown (**B**, solid arrowhead), and concomitant expansion of *Ptep-pnr2* expression into the ventral territory (*n* = 5/5) (**B"**, **B'''**; asterisk). Abbreviations: L3-L4- walking legs 3–4; O1-O2, opisthosomal segments 1–2. Scale bars: 100 μm.

for *Ptep-sog* expression. We found that RNAi against this bona fide gap gene resulted in disruption of *Ptep-sog* in the L1 and L2 segments in *Ptep-Sp6-9* RNAi embryos (S10 Fig). These data suggest that localized abrogation of *sog* in affected segments is a signature of gap segmentation phenotypes. Taken together, the expression dynamics of *Ptep-waist-less* and its RNAi phenotype suggest that *waist-less* acts as a trunk gap gene in the spider *P. tepidariorum*.

## Expression of *Irx* homologs in arachnopulmonate and apulmonate models

The incidence of systemic duplication in arachnopulmonates has spurred inferences of neo-functionalization events that are based on divergence of paralogous gene expression patterns or the discovery of a new function of an arachnopulmonate gene copy [25,26]. However, comparative data from arachnid models with unduplicated genomes are required for testing the robustness of evolutionary scenarios of sub- and neofunctionalization. To understand the evolutionary dynamics of *waist-less* and its sister copies, we generated gene expression surveys of Irx homologs of *P. tepidariorum*, targeting stages that were represented in the DGE analysis (S11–S13 Figs). Given the short length of one of these copies (*Irx2-2*), we were able to generate probes for reliable detection of expression for 4 of the 5 Iroquois genes of *P. tepidariorum*.

*Ptep-Irx1* is uniformly expressed as a thin band in the lateral edge of germband along the entirety of the AP axis with additional expression domains recovered in the head, along the AP

axis as segmentally repeated stripes, and in the prosomal appendages (S14A, S14B, S14A" and S14B" Fig). The expression of *Ptep-Irx1* resembles that of *Ptep-waist-less*, with expression domains of both genes occurring in the lateral-most regions of the germband, head, prosomal appendages, and as segmentally iterated stripes. However, *Ptep-Irx1* differs from *Ptep-waist-less* in that it is not enriched in the opisthosoma, is restricted to a smaller region of the head, and has offset expression domains in the prosomal appendages (S14A–S14A" and S14B–S14B" Fig). Expression of the remaining *Irx* genes was less comparable to *Ptep-waist-less* expression. Expression of *Ptep-Irx2-1* is restricted to the lateral edge of the developing germband, with uniform expression along the entire AP axis and no additional expression domains (S14C and S14C" Fig). *Ptep-Irx3-1* is notably absent from the main body axis and lateral tissues with the expression limited to patches in the pedipalps and walking legs (S14C' and S14C" Fig). *Ptep-Irx2-2* was not assayed as short sequence length prevented accurate design of HCR probes.

These spatial expression patterns generally reflect the bioinformatic predictions of Irx dynamics recovered from DGE analysis in *A. hentzi* (S11–S13 Figs), as exemplified by the elevated expression of *waist-less* in the opisthosomal segments compared to prosomal segments or by the absence of significantly different expression levels between tagmata for *Irx1-1* and *Irx2-1* (S14 Fig). These data suggest that the expression dynamics of Irx homologs are largely conserved between mygalomorph (e.g., *A. hentzi*) and araneomorph (e.g., *P. tepidariorum*) spiders, albeit with the loss of 1 copy in the latter species.

To assess whether enriched opisthosomal expression of *waist-less* is phylogenetically widespread outside of spiders, we assayed the expression of 2 Irx homologs (*Irx2* and *Irx3*) of the apulmonate arachnid *Phalangium opilio* (Fig 6). When the PO boundary is forming (1 opisthosomal segment present: stage 7, sensu [38]), *Popi-Irx2* is expressed in the lateral margin of the germ band, with concentrated expression in the posterior terminus. Expression is not recovered in the head lobes or in the ventral ectoderm (Fig 6A' and 6B'). At stage 9, when the PO boundary is fully formed (3 opisthosomal segments present) expression of *Popi-Irx2* remains comparable to stage 7 with uniform expression in the body wall and enrichment in the posterior terminus (Fig 6C' and 6D'). Notably, the dynamics of *Popi-Irx2* closely resemble the expression of *Ptep-Irx2-1*.

By stage 7, *Popi-Irx3*-positive domains are present in the head lobes and as segmentally iterated patches in the ventral ectoderm (Fig 6A" and 6B"). *Popi-Irx3* shows dynamic expression domains in stage 9 embryos. Expression is maintained in the head and ventral ectoderm, with increased complexity of expression in the central nervous system. New regions of expression are present in the distal portions of the prosomal appendages, to the exception of the chelicera, and show an enrichment of expression in the distal L2 territory. Cheliceral expression is limited to a small area at the base of the appendage. New expression of *Popi-Irx3* is also detected throughout the lateral body wall with strong enrichment in the opisthosoma (Fig 6C" and 6D"). Notably, the dynamics of *Popi-Irx3* closely resemble the expression of *Ptep-waist-less*, suggesting that the association of Irx3 homologs with the opisthosomal boundary is not phylogenetically restricted to spiders or arachnopulmonates.

## Expression of *pannier2* in *Parasteatoda tepidariorum*

The other RNAi experiment that resulted in a morphological phenotype was a GATA family transcription factor. Gene orthology was inferred using a gene tree of GATA sequences. Three spider GATA genes were identified as members of the *pannier* clade, with the highly expressed copy provisionally identified as *Ptep-pnr2* (S15 Fig). We surveyed *pnr2* in wild-type embryos at the developmental stages encompassed by the RNA-seq dataset. At all stages, surveyed *Ptep-pnr2* was expressed in the lateral-most territory of the opisthosoma, which corresponds to the

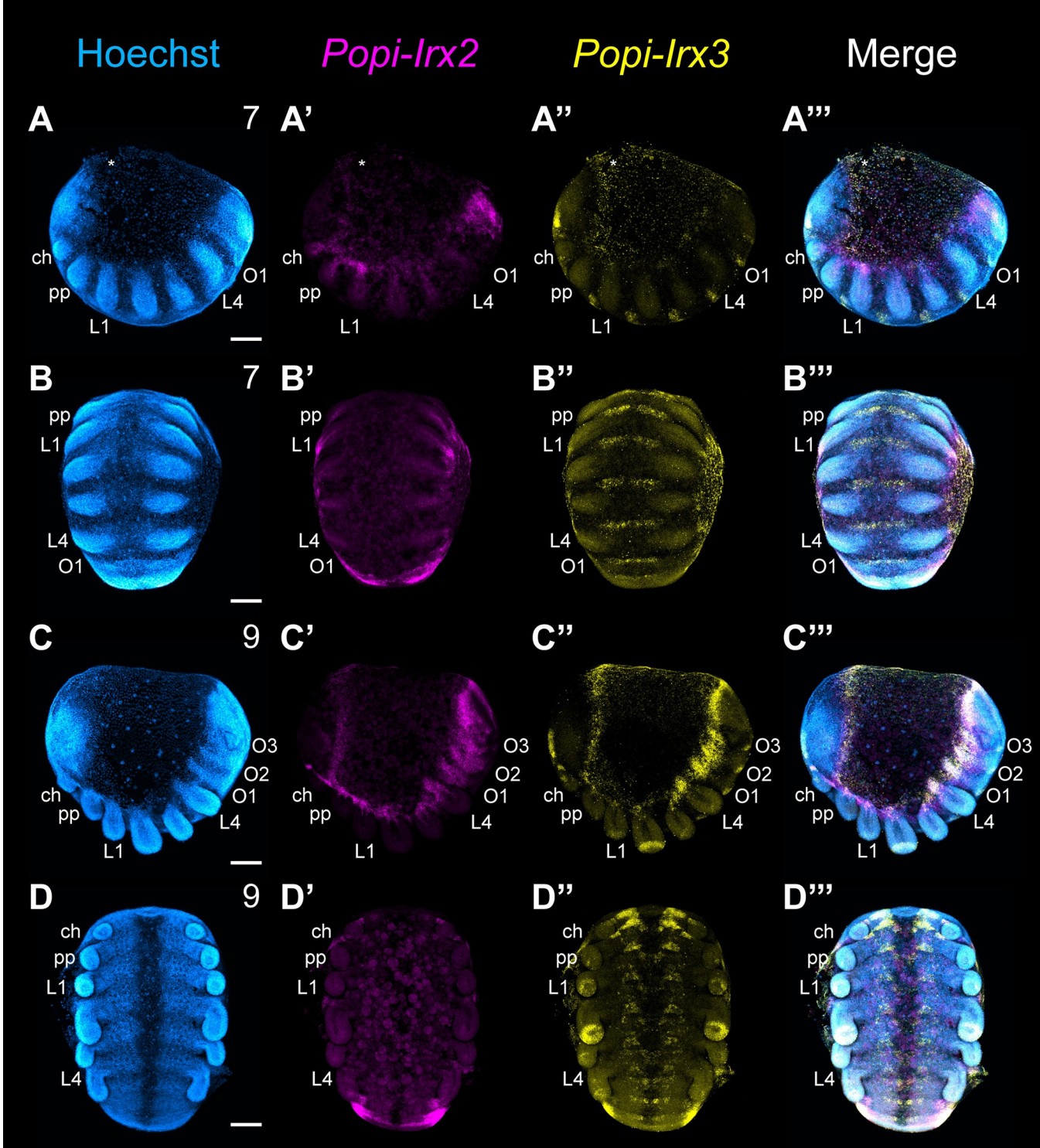

**Fig 6. Expression dynamics of single copy *Irx2* and *Irx3* homologs in the harvestman *P. opilio*.** (**A-B'''**) Expression of *Popi-Irx2* and *Popi-Irx3* in stage 7 embryos. (**C-D'''**) Expression of *Popi-Irx2* and *Popi-Irx3* in stage 9 embryos. Asterisks in A-B''' mark mechanical damage to yolk. Abbreviations as in Fig 2. Scale bars: 100 μm.

dorsal midline upon dorsal closure (S16 Fig). Separate expression domains were recovered in the dorso-lateral margins of the head lobes at stages 9 and 10 (S16A–S16B' Fig). At stage 11, and the initiation of dorsal closure, expression is found throughout the lateral body wall with enrichment and expansion in the dorsally migrating opisthosomal territory (S16C and S16C' Fig).

### Knockdown of *pannier2* results in ectopic dorso-lateral opisthosomal tissue in a spider

The lateral-most edges of the spider germ band correspond to the presumptive dorsal midline, as these 2 margins will fold to enclose the yolk via dorsal closure [31]. Expression of *Ptep-pnr2* in the lateral edges of the developing opisthosoma, with enrichment at stage 11 when inversion and dorsal closure of the embryo begin, suggested a possible function in the patterning of this tagma. Further support for a *Ptep-pnr2* potentially acting in dorsal specification came from the fruit fly *D. melanogaster* wherein *pannier* is necessary for proper dorsal closure of the germ band and is also expressed in the amnioserosa (extraembryonic membrane; absent in chelicerates) [39–44].

Investigating the function of *Ptep-pnr2* using maternal RNAi, we found a small proportion of *Ptep-pnr2* RNAi embryos displayed ectopic opisthosomal tissue resulting in a smaller proportion of extraembryonic territory in affected embryos at developmental stages associated with the beginnings of inversion and dorsal closure ($n = 58/646$) (Fig 7B), as well as abnormal pouches resembling ectopic neuromeres (Fig 7C and 7D). Additionally, *Ptep-pnr2* RNAi embryos exhibited ectopic *Ptep-sog* expression in the dorsal margin of the opisthosoma (Fig 7C' and 7D'), suggesting that *Ptep-pnr2* represses ventral identity. These data are consistent with the interpretation that *Ptep-pnr2* RNAi embryos exhibit a dorso-ventral defect, wherein the dorsal midline takes on ventral identity in the absence of *Ptep-pnr2*. Due to the low penetrance of *Ptep-pnr2* RNAi, the function of *Ptep-pnr2* was not pursued further.

## Discussion

### The Iroquois homolog *waist-less* is a spider gap segmentation gene

Comparative investigations of arthropod body plan evolution have historically focused on various aspects of morphogenesis, such as anteroposterior segmentation, neurogenesis, regionalization of body axes, and germ cell specification. Candidate gene approaches in spiders have featured prominently in such investigations, with *P. tepidariorum* serving as the leading model system representing Chelicerata. In some cases, the magnitude of the phylogenetic distance between chelicerates and insects has limited the informativeness of candidate gene suites that were established from the fruit fly literature. A separate challenge for an insect model-derived candidate gene approach is the evolution of taxon-restricted genes, as exemplified by the subdivision of *araucan* and *caupolican* (restricted to a derived group of dipterans) and by the abundance of gene duplicates resulting from WGD in groups like spiders. Here, we developed a tissue-specific transcriptomic profile of appendage-bearing segments in a large-bodied spider to circumvent these hurdles. Profiling for, and functional screening of, genes highly expressed in the spider posterior tagma resulted in the identification of *waist-less*, an Iroquois gene whose ortholog has been lost in the common ancestor of Pancrustacea. The high level of expression posterior to the PO boundary for *waist-less* and *pannier2*, as well as their respective roles in territory-specific segmental and dorso-ventral patterning, accorded with bioinformatic predictions of the DGE analysis.

The phenotypic spectrum incurred by RNAi against *waist-less* was unexpected for an Iroquois homolog. The sparse existing data for Iroquois family genes in chelicerate taxa have

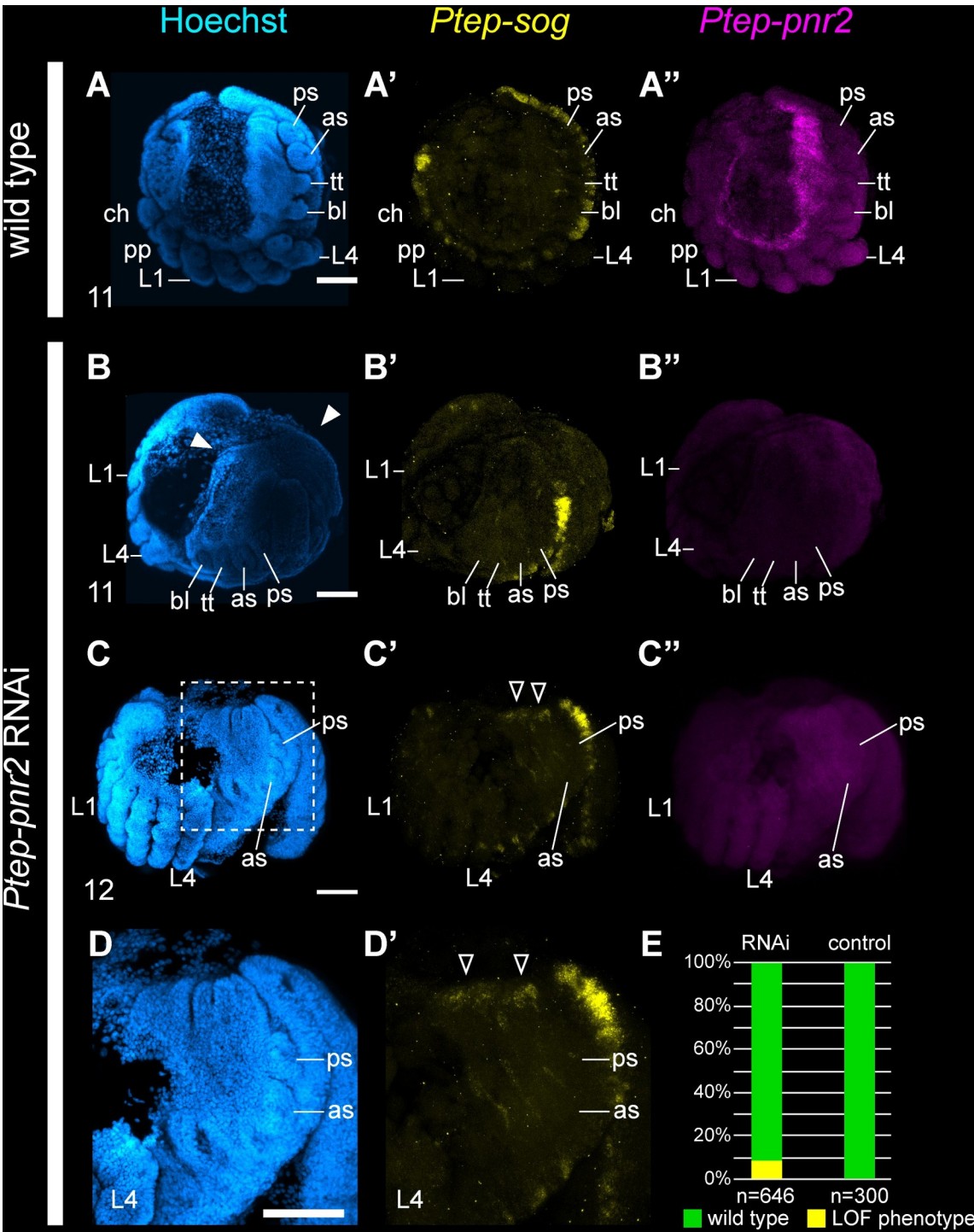

**Fig 7. RNAi against *Ptep-pnr2* results in ectopic tissue formation in the opisthosoma and disrupts expression of *Ptep-sog*. (A-A")** Wild-type embryos at stage 11 (inversion) show *Ptep-sog* expression restricted to the ventral midline (**A'**) and *Ptep-pnr2* is expressed in the lateral margins of the germ band with strongest expression concentrated in the opisthosoma and migrating tissues (**A"**) (*n* = 9/9). (**B-C"**) *Ptep-pnr2* loss-of function phenotypes exhibit expansion of the lateral opisthosomal territory (**B**, solid arrowheads). In later stages, ectopic expression of *Ptep-sog* was detected in the dorsal midline of the opisthosoma in *Ptep-pnr2* RNAi embryos (*n* = 4/7) (**C'**, **D'**, open arrowheads). Expression of *Ptep-pnr2* was disrupted and indistinguishable from background in *Ptep-pnr2* RNAi embryos (**B"**, **C"**). (**D, D'**) Magnification of C-C' corresponding to the region outlined in C. (**E**) Phenotypic distribution of *Ptep-pnr2* RNAi and control embryos. Abbreviations as in Figs 2 and 3. Scale bars: 100 µm. The data underlying the graphs shown in the figure can be found in S1 Data.

encompassed only bioinformatic assays and whole mount gene expression surveys, leaving the function of this gene family unexplored in non-insect Arthropoda. Expression patterns of spider Iroquois family genes were previously interpreted to mean that these paralogs have undergone subfunctionalization after duplication in spider development, upon comparison to the expression pattern in a non-arachnopulmonate arachnid lineage (arachnids lacking a WGD event, such as harvestmen) [25]. However, that previous survey reported only one of the 3 Iroquois genes in the harvestman, and only 4 of the 5 in the spider *P. tepidariorum*. In addition, the expression dynamics of *waist-less* transcripts in single-cell RNA-seq datasets had previously been interpreted to mean that this spider gene played a role in antero-posterior segmentation and/or neural development [33].

By contrast, in the fruit fly *D. melanogaster*, the 2 homologs of *waist-less* (cyclorraphan fly-specific duplicates *araucan* and *caupolican*) are broadly pleiotropic, acting in multiple contexts that span dorso-lateral patterning of the body wall; heart, eye, and muscle development; extra-embryonic tissue specification; and neurogenesis and sensory structure development [45–51]. In many of these functional contexts, Irx copies of the *ara/caup* group have been demonstrated to serve redundant functions, including dorso-ventral patterning [47]. Nevertheless, a role in regionalized tissue maintenance (i.e., the discontinuous germ bands in loss-of-function phenotypes that were found in this study) and segmentation was not known for any arthropod Iroquois homolog.

Our available data suggest that *waist-less* acts as a gap gene at the boundary of the 2 spider tagmata. The discovery of the gap gene that patterns the posterior prosoma and the anterior opisthosoma fills a long-standing gap in spider developmental genetics. The identification of spider gap genes has proven difficult, especially in regard to the PO boundary (Fig 8) [16,30,36,52]. Previous work on spider segmentation revealed a canonical gap segmentation function for the spider homolog of *hunchback*, typically resulting in the deletion of the first and second walking legs in RNAi experiments (Fig 8A) [52]. Other classical gap genes from the insect literature like *Krüppel*, *knirps*, and *giant* are either not expressed comparably in spiders [6,14] or do not exhibit a gap phenotype in RNAi experiments in our hands. Redundant roles in patterning this same territory were subsequently established for the distal limb-patterning genes *Distal-less* and *Sp6-9*, exemplifying co-option of an appendage-patterning regulatory cassette to fulfill a segmentation function (Fig 8A) [30,36,53]. Broader segmentation phenotypes have been elicited by targeting components of the Wnt pathway or the Sox family of transcription factors [6,16,29], but the phenotypic spectra resulting from these experiments are not directly comparable to a trunk gap segmentation phenotype.

Our results suggest that the function of spider *waist-less* is analogous to the function of *Krüppel* in insect models, wherein this classical gap gene patterns segments of the posterior thorax and anterior abdomen (Fig 8B) [54,55], whereas the more anteriorly acting *hunchback* patterns the posterior gnathos through the thorax [56]. As the present study was focused on characterizing the phenotype of *waist-less*, we did not examine the regulatory relationships of *hunchback* and *waist-less* in the spider here, but we anticipate these to be fertile ground for future investigations of arachnid segmentation.

## Association of *Iroquois3* with the prosoma-opisthosoma boundary predates the WGD of arachnopulmonates

Investigating the genetic architecture of the spider body plan highlights the challenges of the candidate gene approach in emerging model systems; classical arthropod models (holometabolous insects) lack the pedicel, as well as other taxon-specific structures of interest. This has precluded developmental genetic investigations of iconic arachnid organs like spider venom

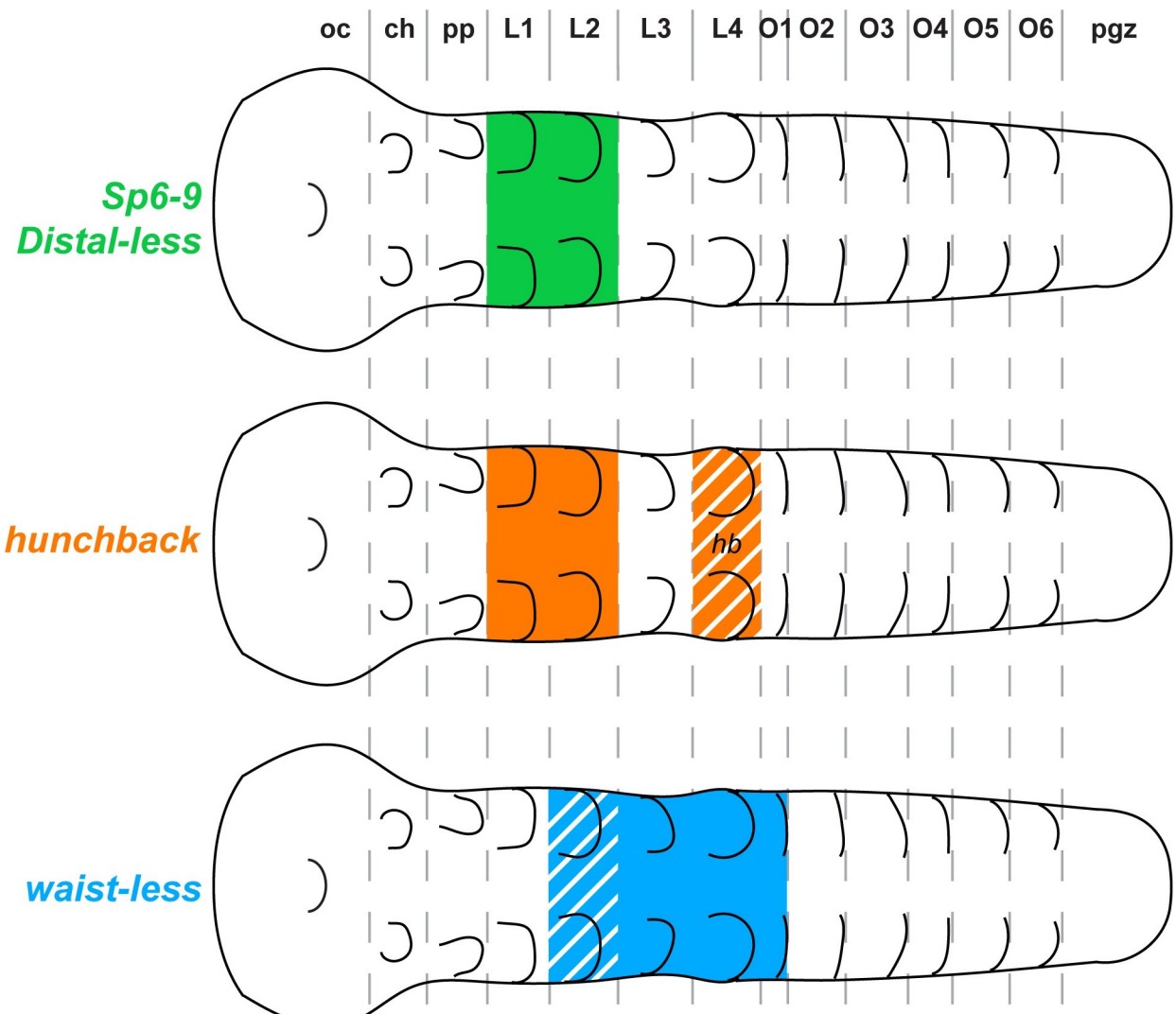

**Fig 8. Summary of arachnid gap genes.** Previously reported spider gap segmentation phenotypes are restricted to the prosoma and comprise a small number of genes: *Ptep-Sp6-9*, *Ptep-Dll*, *Ptep-hb*, and *Ptep-Sox21b-1*. *Ptep-waist-less* constitutes a newly discovered gap gene that patterns the PO boundary. *Ptep-Sox21b-1* is pleiotropic and omitted from this figure for simplicity.

glands, silk spigots, and fangs. The RNA-seq datasets established herein for limb bud-bearing territories of embryonic spiders, spanning developmental stages most salient to development of posterior appendages, are anticipated to guide future investigations of book lung and (spider) spinneret development, 2 arachnid organs that have prompted prolonged debates over evolutionary origins and serial homology [57–59].

One potential limitation of spiders as model systems for study of chelicerate developmental biology is the incidence of WGD in the common ancestor of Arachnopulmonata, as the ancestral condition for Chelicerata is unambiguously an unduplicated genome [20,60,61]. In the context of the present study, the function of *waist-less* is of interest from the perspective of body plan evolution, but *waist-less* is itself a duplicated copy; groups like mites, ticks, and harvestmen possess only 1 homolog of *Iroquois3*. It was therefore not clear whether the function of *waist-less* reflects a dynamic conserved across Chelicerata, or whether it represents an arachnopulmonate-specific novelty. Here, we found that the single-copy homolog of *Iroquois3* is

expressed comparably to *waist-less* in embryos of the harvestman *Phalangium opilio*. These data suggest that an association between *Iroquois3* and the PO boundary predates the divergence of arachnopulmonates and that the function of *waist-less* does not represent an evolutionary novelty within chelicerates, much less a spider-specific innovation. A future test of this hypothesis should target functional data in the harvestman, with the prediction that disrupting the single-copy *Iroquois3* homolog will result in the same gap segmentation phenotype in this species.

The increasing availability of expanded genomic resources for chelicerate groups may facilitate broader tests of how arachnopulmonate copies diverge as a function of phylogenetic distance. Specifically, the recently established availability of developmental genetic resources for non-spider arachnopulmonates like scorpions [22] and whip spiders [62] may enable investigation of whether Iroquois duplicates faithfully retain expression domains as a function of orthology or whether they exhibit developmental system drift. As a corollary, investigating the activity of *Iroquois3* in chelicerate taxa that have undergone modifications of the PO boundary may aid in testing the inference of opisthosoma-specific activity of these 2 genes. Specifically, comparative data from Pycnogonida (sea spiders, which retain only a rudiment of the opisthosoma) and mites of the superfamily Eriophyoidea (gall mites, which have undergone dramatic reduction of the L3 and L4 segments) may aid in understanding how tagmata evolve in the context of segmental reduction [63,64].

## Methods

### Field collection, sequencing, and differential gene expression analyses of tarantula embryos

Field collection protocols for *Aphonopelma hentzi* (Araneae: Theraphosidae) embryos and laboratory protocols for care were previously described by Setton and colleagues [27]. Embryos used to generate the reference developmental transcriptome were stored in TRIzol TRI Reagent (Ambion Life Technologies, Waltham, Massachusetts, United States of America) prior to RNA extraction, following manufacturer's protocols. Library preparation and stranded mRNA sequencing were performed at the University of Wisconsin-Madison Biotechnology Center on an Illumina HiSeq 2500 platform with 2x100 PE reads. The transcriptome spans developmental stages 9.1 to 13 (following [27,31]) to juveniles (first and second instar posthatching). This resource is available under accession numbers NCBI SRR13605914 and SRR13605915 [20].

Three sets of biological replicates for each tarantula appendage type along with tissue from a portion of the corresponding segment that was directly attached to the appendage (i.e., associated body wall tissue) were dissected at 3 different time points during embryogenesis (stages 9, 10, and 11 after Setton and colleagues' study [27]). Each experimental sample contained appendages from multiple individuals from the same clutch ($n$ = 3 to 7 samples per appendage type). Total RNA was extracted using TRIzol TRI Reagent (Ambion Life Technologies, Waltham, Massachusetts, USA), following the manufacturer's protocol. Libraries were prepared for sequencing using standard protocols for the Illumina NovaSeq 6000 platform with a 2x150 PE sequencing strategy (stages 9 and 10) or the Illumina HiSeq 2500 platform with 1x100 SE sequencing strategy (stage 11). Multiplexing was designed to recover an expected 15M reads per library for stages 9 and 10, and 12M reads per library for stage 11. Adaptor removal and quality trimming was conducted using Trimmomatic v 0.35 [65] prior to analysis. The majority of samples sequenced using the PE sequencing strategy are represented by 2 libraries for each sequencing direction. In these instances, like libraries were concatenated after adaptor

removal and quality trimming (Trimmomatic v 0.35). All unconcatenated raw read files have been accessioned to NCBI Sequence Read Archive (SRA) as BioProject PRJNA1105064.

Reads were mapped to the *A. hentzi* developmental transcriptome using the density of reads mapped to the transcriptome as a proxy for transcript abundance, as implemented by salmon v. 0.9.1 under default parameters [66]. DGE analysis was performed using DESeq2 v. 1.14.1 [67]. Both programs were run under standard parameters, with cutoff values for identification of differentially expressed genes in DESeq2 set to a *p*-value of $\leq 0.05$ and an LFC $\geq 1$. These cutoffs were applied to both all-by-all and pairwise comparisons for the identification of differentially expressed genes.

## Gene tree analysis and orthology inference

BLAST and BLASTp searches were used to determine the identities of transcripts identified in DGE datasets. Orthology of *A. hentzi Iroquois* homologs was determined using previously published *P. tepidariorum Irx* sequences as queries [26,33] for tBLASTn searches, and hits with e-values $< 10^{-5}$ were retained. All putative orthologs were verified using reciprocal BLAST searches. Multiple sequence alignment was conducted de novo with MAFFT v.7 with default parameters [68]. Identification of *A. hentzi pannier* homologs was determined using previously published *D. melanogaster pannier* and *grain* sequences as queries for tBLASTn searches, and hits with e-values $< 10^{-5}$ were retained. Vertebrate sequences included the GATA1-3 (*grain*) group and GATA4-6 (*pnr*) group and were taken from a previously published study [69]. All putative orthologs were verified via reciprocal BLAST searches, as with Iroquois orthologs. Multiple sequence alignment was conducted de novo with CLUSTAL Omega [70].

Phylogenetic reconstruction of Iroquois amino acid alignments consisted of maximum likelihood analysis with IQ-TREE, with automated model selection (-m MFP; chosen model: LG +G4) and 1,000 ultrafast bootstrap resampling replicates [71]. Chelicerate sequences were pulled from previously published genome or transcriptome assemblies available on GenBank, and insect sequences were added from a previous work on the Iroquois gene family [32]. Phylogenetic reconstruction of *pannier* amino acid alignments consisted of maximum likelihood analysis with IQ-TREE, with automated model selection (-m MFP; chosen model: DCMut+F +R5) and 1,000 ultrafast bootstrap resampling replicates. All alignments, annotated tree files, and log files are available upon request.

## Cloning of orthologs and probe synthesis

Fragments of *Ptep-waist-less* and *Ptep-pnr2* were amplified using standard PCR protocols and cloned using the TOPO TA Cloning Kit using One Shot Top10 chemically competent *Escherichia coli* (Invitrogen, Carlsbad, California, USA) following the manufacturer's protocol, and their PCR product identities were verified via sequencing with M13 universal primers. All gene-specific primer sequences are provided in S4 Table. Upon completion of probe synthesis, the presence of the target sequence was checked using gel electrophoresis.

## House spider embryo collection, fixation, in situ hybridization, and imaging

Animals were maintained, and embryos fixed and assayed for gene expression, following established or minimally modified protocols for colorimetric in situ hybridization, as detailed previously [27,30,72]. PCRs for generating riboprobe templates, synthesis of DIG-labeled probes, and preservation of embryos all followed recently detailed procedures [29,30]. Whole mount images were taken using a Nikon SMZ25 fluorescence stereomicroscope mounted with a DS-Fi2 digital color camera driven by Nikon Elements software.

For hybridization chain reaction (HCR) gene expression assays, probes were designed separately for each gene using an open-source probe design platform [73] with standard parameters and 20 probe pairs per gene returned; *Ptep*-sog was designed with the delay parameter set to 50, and the number of probe pairs was set to 30. Expression for *Ptep-Irx2-2* could not be surveyed using HCR due to the short sequence length (747 bp). For instances where genes with regions of high sequence similarity were multiplexed into a single probe, the regions of highly similar sequence were identified using sequence alignments and then masked in Aliview v.1.28 prior to probe design [74]. All probes were designed to span a maximal amount of ORF and minimal UTR (S5 Table).

For HCR, embryos of *P. tepidariorum* were fixed by dechorionation in 50% bleach solution (Clorox brand) and fixed in a 3.2% paraformaldehyde (PFA) solution in PBS for 35 min. Vitelline membranes were manually removed using fine forceps during the fixation in PFA solution. Embryos were washed in PBS-Tween20 several times and serially dehydrated into 100% ethanol for storage at −20˚C. The procedures for HCR, and all solutions therein, constitute minor modifications of a recently published protocol [75]. For spiders, we lowered the amount of probe hybridization solution to 148 μL and added in probe stocks at 2× to 4× suggested concentration (1.6 μL to 3.2 μL probe per gene). Confocal imaging was conducted on a Zeiss LSM710 confocal microscope driven by Zen software.

## Double-stranded RNA synthesis and maternal RNA interference

dsRNA was synthesized following the manufacturer's protocol using a MEGAscript T7 kit (Ambion/Life Technologies, Grand Island, New York, USA) from amplified PCR product. dsRNA quality was checked, and concentration adjusted to 2.5 μg/μl. For *Ptep-waist-less*, RNAi was performed with 20 μg of dsRNA of a 978-bp fragment, delivered 4 times over 8 days (with injection every other day) to 32 virgin females, with 22 surviving to laying the second cocoon. Of these 22 females injected with *Ptep-waist-less* dsRNA, 13 produced at least 1 cocoon of embryos with phenotypes; embryos were collected from cocoons 2 to 5 as previously described [29,30]. Negative controls were injected with an equal volume of deionized water, following established protocols in spiders [17]; 12 females were injected, with 7 laying beyond cocoon 2. To rule out off-target effects of RNAi, we performed gene silencing using 2 non-overlapping fragments of *Ptep-waist-less* (473 bp and 405 bp fragments), delivered at the same concentration and under the same timeline as the 978-bp fragment experiment, and assessed the resulting phenotypic spectra to confirm identical phenotypes. Counts of phenotypes were obtained from a randomly selected group of embryos spanning clutches 3 to 5 of multiple females, for both RNAi and negative control experiments.

For *Ptep-pnr2* RNAi, dsRNA was injected at a concentration of 4 μg/μl for a total of 32 μg administered over 8 days, following optimization of dsRNA delivery for this gene. Three virgin females were injected 4 times as in the *Ptep-waist-less* experiments, with another 2 females injected as negative controls. Three out of 4 females laid egg sacs, and embryos were collected from cocoons 2 to 5 as previously described [29,30]. Counts of phenotypes were obtained from a randomly selected group of embryos spanning clutches 3 to 5 of multiple females, for both RNAi and negative controls experiments.

RNAi against the gap gene *Sp6-9* reproduced the procedures described above, with identical amplicons for dsRNA synthesis as reported in previous studies [30]. Briefly, 2 to 3 females were injected every other day to deliver 20 μg of dsRNA. Embryos exhibited gap phenotypes were sampled from the second and third egg clutches at the limb bud stage and fixed for colorimetric in situ hybridization, following procedures described above.

## Supporting information

**S1 Data. Raw data underlying the graphs shown in the figures.**
(XLSX)

**S2 Data. Raw and normalized read counts for *Aphonopelma hentzi* embryonic appendages, from stage 9.**
(XLSX)

**S3 Data. Raw and normalized read counts for *Aphonopelma hentzi* embryonic appendages, from stage 10.**
(XLSX)

**S4 Data. Raw and normalized read counts for *Aphonopelma hentzi* embryonic appendages, from stage 11.**
(XLSX)

**S1 Table. List of genes screened via RNAi in *Parasteatoda tepidariorum*.**
(DOCX)

**S2 Table. List of RNAi candidate gene identifiers in *Aphonopelma hentzi* and *Parasteatoda tepidariorum* transcriptomic and genomic resources.**
(DOCX)

**S3 Table. List of genomes referenced for synteny analysis.**
(DOCX)

**S4 Table. List of primer sequences used for gene cloning and/or riboprobe synthesis.**
(DOCX)

**S5 Table. List of HCR probe sequences.**
(DOCX)

**S1 Fig. Transcriptional profiles and principal components analysis of territory-specific gene expression in embryos of the tarantula *Aphonopelma hentzi*.** From left to right: stage 9, stage 10, and stage 11. Arrowhead indicates tarantula ortholog of *waist-less*. Complete dataset for heatmaps is provided in S2–S4 Data.
(ZIP)

**S2 Fig. DGE profiles of candidate genes highly expressed in prosomal regions of developing *A. hentzi* embryos. Expression levels are shown by RNA-seq library (tissue type) in TPM.** Prosoma-enriched genes were not further pursued after initial RNAi screens in *P. tepidariorum* due to high mortality (*SPDEF*, *Ahen-TRINITY_DN6222_c0_g1*, *spaetzle*) or no discernable phenotype (*Sox8*, *Pax9-1*, *Ahen-TRINITY_DN3695_c1_g1*, *piopio*). For orthologous gene identifications of uncharacterized transcripts between *A. hentzi* and *P. tepidariorum*, see S2 Table. Complete dataset is provided in S2–S4 Data. The data underlying the graphs shown in the figure can be found in S1 Data. DGE, differential gene expression; RNAi, RNA interference; TPM, transcripts per million.
(TIF)

**S3 Fig. DGE profiles of candidate genes highly expressed in opisthosomal regions of developing *A. hentzi* embryos.** Expression levels are shown by RNA-seq library (tissue type) in TPM. RNAi screens against opisthosoma-enriched genes in *P. tepidariorum* resulted in 2 phenotypes (*pnr2*, *waist-less*). The remaining candidates resulted in high mortality (*Mab21-1*) and/or no discernable phenotype (*Mab21-1*, *biniou*, *Hand2-2*). For orthologous gene

identifications of uncharacterized transcripts between *A. hentzi* and *P. tepidariorum*, see S2 Table. Complete dataset is provided in S2 and S3 Data. The data underlying the graphs shown in the figure can be found in S1 Data. DGE, differential gene expression; RNAi, RNA interference; TPM, transcripts per million.
(TIF)

**S4 Fig. Maximum likelihood tree topology of panarthropod Iroquois homologs.** Colors correspond to major phylogenetic lineages. Numbers on nodes correspond to bootstrap resampling frequencies.
(TIF)

**S5 Fig. Wild-type expression of *waist-less* during embryogenesis of the cobweb spider *Parasteatoda tepidariorum*.** Panel A constitutes a brightfield-only image of DIG-labeled in situ hybridization for *Ptep-waist-less*; panels B-W constitute merged images of Hoechst and DIG-labeled in situ hybridization for *Ptep-waist-less*. Abbreviations as in Fig 2. Scale bar: 100 μm.
(TIF)

**S6 Fig. Validation of RNAi using colorimetric in situ hybridization.** Upper row: negative control embryo showing wild-type expression of *Ptep-waist-less*. Lower row: RNAi embryo showing diminution of *Ptep-waist-less* expression and abnormal development of germ band in the territory abutting the posterior growth zone. Scale bar: 100 μm.
(ZIP)

**S7 Fig. Effects of *Ptep-waist-less* RNAi in late stages of embryogenesis and at hatching.** (**A**-**C**) Negative control embryos exhibiting wild-type development. (**D**) Stage 11 Class I RNAi embryo exhibiting discontinuous germ band and aberrant disposition of opisthosoma. (**E**) Stage 13 Class II RNAi embryo exhibiting anomalous development of the pedicel territory and constriction of germ band due to missing tissue between tagmata. (**F**) Postembryo from RNAi experiment with mosaic phenotype, exhibiting loss of L4 (asterisk) and smaller prosoma on affected side (note position of dotted line in midline of the prosoma). Scale bar: 100 μm.
(TIF)

**S8 Fig. Wild-type expression of *Ptep-pnr2* shown in context with *Ptep-waist-less* and the ventral midline marker *Ptep-sog*.** Across all 3 stages surveyed for DGE in *A. hentzi*, *Ptep-pnr2* is expressed in the lateral edge of the germ band, which will become the dorsal part of the spider. In accordance with the DGE data, the strongest expression of *Ptep-pnr2* is in the opisthosoma ($n = 12/12$). Abbreviations as in Fig 2. Scale bars: 100 μm.
(TIF)

**S9 Fig. Expression of *Ptep-pnr2* in a Class I *Ptep-waist-less* loss-of-function phenotype is disrupted in the opisthosoma.** In *Ptep-waist-less* RNAi embryos, the expression of *Ptep-pnr2* is no longer cleanly defined in the lateral edge of the opisthosoma and becomes blurred ($n = 6/9$; wild type $n = 12/12$). Abbreviations as in Fig 2. Scale bars: 100 μm.
(TIF)

**S10 Fig. Expression of the ventral midline marker *sog* is disrupted in *Ptep-Sp6-9* RNAi embryos.** (**A**) Expression of *Ptep-sog* is continuous throughout the ventral midline in wild-type embryos. (**B**) RNAi against *Ptep-Sp6-9* results in a gap segmentation phenotype concomitant with the loss of *Ptep-sog* expression in the affected regions. (**B'**) Same embryo as in B, with Hoechst counterstain. Abbreviations as in Fig 2. Scale bar: 100 μm.
(TIF)

**S11 Fig. DGE profile of *Iroquois* homologs in *A. hentzi* embryos at stage 9.** (**A**) Expression levels for each Iroquois homolog by RNA-seq library (tissue type) in TPM. (**B**) Individual expression profiles of homologs by tissue type (magnified from panel A) show *Ahen-waist-less* is not comparably expressed to *Ahen-Irx3*-1 or other Iroquois homologs. Transcripts of *Ahen-waist-less* are highly enriched in RNA-seq libraries of opisthosomal tissue, to the exclusion of all prosomal regions sampled. Complete dataset is provided in S2 Data. The data underlying the graphs shown in the figure can be found in S1 Data. bl, book lung; ch, chelicera; lb, labrum; L1, first walking leg; pp, pedipalp; sp, spinnerets.
(TIF)

**S12 Fig. DGE profile of *Iroquois* homologs in *A. hentzi* embryos at stage 10.** (**A**) Expression levels for each *Iroquois* homolog by RNA-seq library (tissue type) in TPM. (**B**) Individual expression profiles of homologs by tissue type (magnified from panel A) show *Ahen-waist-less* is not comparably expressed to *Ahen-Irx3*-1 or other *Iroquois* homologs. Transcripts of *Ahen-waist-less* are enriched in RNA-seq libraries of opisthosomal tissue. Complete dataset is provided in S3 Data. The data underlying the graphs shown in the figure can be found in S1 Data. as, anterior spinneret; bl, book lung; ch, chelicera; lb, labrum; L1, first walking leg; pp, pedipalp; ps, posterior spinneret.
(TIF)

**S13 Fig. DGE profile of *Iroquois* homologs in *A. hentzi* embryos at stage 11.** (**A**) Expression levels for each *Iroquois* homolog by RNA-seq library (tissue type) in TPM. (**B**) Individual expression profiles of homologs by tissue type (magnified from panel A) show *Ahen-waist-less* is not comparably expressed to *Ahen-Irx3*-1 or other *Iroquois* homologs. Transcripts of *Ahen-waist-less* are enriched in RNA-seq libraries of opisthosomal tissue. Complete dataset is provided in S4 Data. The data underlying the graphs shown in the figure can be found in S1 Data. as, anterior spinneret; bl, book lung; ch, chelicera; lb, labrum; L1, first walking leg; pp, pedipalp; ps, posterior spinneret.
(TIF)

**S14 Fig. Expression of *Iroquois* homologs in *P. tepidariorum*.** (**A**-**B'''**) *Ptep-Irx1* and *Ptep-waist-less* are similarly expressed but have distinct expression domains. *Ptep-waist-less* is enriched in body wall tissue of the opisthosoma, has a broader expression territory in the head, and has additional expression domains in the legs, as compared to *Ptep-Irx1* (**A''**, **B''**). (**C-C'''**) *Ptep-Irx2-1* and *Ptep-Irx3-1* are not comparably expressed to *Ptep-waist-less*. *Ptep-Irx2-1* is restricted to a uniform band of expression along the lateral margin of the germ band, with slight protrusions into the proximal-most regions of developing appendages (**C**). Expression of *Ptep-Irx3-1* is restricted to 2 non-overlapping expression domains in the developing legs and pedipalps, 1 distal and 1 medial. *Ptep-Irx3-1* is notably absent from the chelicera (**C'**). Asterisks in C-C''' mark mechanical damage to yolk; note autofluorescence in C' and C'' for *Ptep-Irx3-1*. Abbreviations as in Fig 2. Scale bars: 100 μm.
(TIF)

**S15 Fig. Maximum likelihood tree topology of GATA homologs.** Colors correspond to previously identified paralogs of GATA. Numbers on nodes correspond to bootstrap resampling frequencies. Boldface text indicates *pannier2* copies of spiders.
(TIF)

**S16 Fig. Wild-type expression of *pnr2* in the cobweb spider *Parasteatoda tepidariorum* at stages targeted by DGE analysis.** All panels constitute split channel images of the same embryo for Hoechst and HCR in situ hybridization for *Ptep-pnr2* ($n$ = 12/12). Abbreviations

as in Fig 2. Scale bar: 100 μm.
(TIF)

## Acknowledgments

Commentary from an anonymous reviewer and A. Matteen Rafiqi significantly refined the ideas and experiments presented in this work. Confocal microscopy was performed at the Newcomb Imaging Center, Department of Botany, University of Wisconsin–Madison. Guilherme Gainett assisted with the pipeline for DGE analyses.

## Author Contributions

**Conceptualization:** Emily V. W. Setton, Prashant P. Sharma.

**Data curation:** Emily V. W. Setton, Jesús A. Ballesteros.

**Formal analysis:** Emily V. W. Setton, Jesús A. Ballesteros.

**Funding acquisition:** Emily V. W. Setton, Prashant P. Sharma.

**Investigation:** Emily V. W. Setton, Pola O. Blaszczyk, Benjamin C. Klementz, Prashant P. Sharma.

**Methodology:** Emily V. W. Setton.

**Project administration:** Emily V. W. Setton, Prashant P. Sharma.

**Resources:** Emily V. W. Setton, Prashant P. Sharma.

**Supervision:** Jesús A. Ballesteros, Prashant P. Sharma.

**Validation:** Emily V. W. Setton, Prashant P. Sharma.

**Visualization:** Emily V. W. Setton, Pola O. Blaszczyk, Prashant P. Sharma.

**Writing – original draft:** Emily V. W. Setton.

**Writing – review & editing:** Emily V. W. Setton, Prashant P. Sharma.

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
