## [Editor Report · Decision Letter 0]

15 Aug 2023

Dear Dr Setton, 

Thank you for submitting your manuscript entitled "A taxon-restricted duplicate of Iroquois3 is required for patterning the spider waist" for consideration as a Research Article by PLOS Biology.

Your manuscript has now been evaluated by the PLOS Biology editorial staff as well as by an academic editor with relevant expertise and I am writing to let you know that we would like to send your submission out for external peer review.

Once your full submission is complete, your paper will undergo a series of checks in preparation for peer review. After your manuscript has passed the checks it will be sent out for review. To provide the metadata for your submission, please Login to Editorial Manager (https://www.editorialmanager.com/pbiology) within two working days, i.e. by Aug 17 2023 11:59PM.

Kind regards,

Ines

--

Ines Alvarez-Garcia, PhD

Senior Editor

PLOS Biology

---

## [Decision Letter · Decision Letter 1]

5 Oct 2023

Dear Dr Setton,

Thank you for your patience while your manuscript entitled "A taxon-restricted duplicate of Iroquois3 is required for patterning the spider waist" was peer-reviewed at PLOS Biology. It has now been evaluated by the PLOS Biology editors, an Academic Editor with relevant expertise, and by two independent reviewers. 

The reviews are attached below. As you will see, both reviewers are positive and find the conclusions interesting, but they also raise some concerns that would need to be addressed before we consider the manuscript for publication. Reviewer 1 thinks that the methods should be improved, as they are incomplete, and that you should provide expression profiles of the 16 genes as it is shown for the Irx genes, among other issues. Reviewer 2 mainly would like you to clarify several points.

In light of the reviews and discussion with the Academic Editor, we would like to invite you to revise the work to thoroughly address the reviewers' reports.

Given the extent of revision needed, we cannot make a decision about publication until we have seen the revised manuscript and your response to the reviewers' comments. Your revised manuscript is likely to be sent for further evaluation by all or a subset of the reviewers.

**IMPORTANT - SUBMITTING YOUR REVISION**

3. Resubmission Checklist

a) *PLOS Data Policy*

b) *Published Peer Review*

Sincerely,

Ines

--

Ines Alvarez-Garcia, PhD

Senior Editor

PLOS Biology

Reviewers' comments

Rev. 1:

This paper describes the identification of the spider waist-less gene, encoding an Irx3 homeodomain protein, which functions in the formation of the boundary region between the prosoma and opisthosoma, two tagmata characteristic of chelicerates. The authors performed segment-specific RNA-seq and differential gene expression analyses in the large spider, Aphonopelma hentzi, followed by functional screening of orthologous genes in another spider, Parasteatoda tepidariorum. These analyses allowed them to identify the waist-less gene whose maternal RNAi in P. tepidariorum caused severe damage to the formation of the germ band, the loss or malformation of segments at and near the boundary between the prosoma and opisthosoma and discontinuity of the germ band at this boundary region. In the subsequent analysis of knockdown phenotypes, they found that the expression of the dorso-ventral patterning genes, sog and pannier2 was affected; they speculated the involvement of the waist-less gene in dorso-ventral patterning in the boundary region. The authors performed a molecular phylogenetic analysis of Irx genes to show that the Irx3 group, which includes the waist-less gene, is not present in hexapods and crustaceans. This indicates that the formation of the body boundary between the characteristic tagmata of chelicerates is linked to the function of a gene present only in chelicerates and myriapods. Additionally, they knocked-down pannier2 in P. tepidariorum to show an aberrant dorso-ventral pattern in a small percentage of embryos.

This study is valuable considering that the waist-less gene was identified based only on the spider transcriptomes and not through candidate screening. Irx3 homologs are not present in the well-studied insects, and the phenotypes of the waist-less knockdown embryos detected in this study are novel. However, in some cases, the methods were not clearly stated, and the presentation of the data was not appropriate. The phenotypes of the knockdown embryos should also be examined at earlier stages of development. There are several major concerns that should be addressed.

Major points:

1. Further information regarding the tissue-specific RNA-seq methods and subsequent DGE analyses should be provided. Was total RNA extracted from the main body, lateral-dorsal parts of the body, or appendages? (p5, l09 or p20, 490.) What is the expression pattern of the identified genes in the embryo? Were they expressed region specifically? Brief statements summarizing the DGE analysis are needed. With respect to the waist-less gene, the expression does not seem clearly different among the segments of the opisthosoma and the prosoma in the stained embryos (fig. S3). In the RNA-seq data (fig. S11-13), the expression levels appear to be correlated with the expression in the appendages. Further comments and discussion on these points are needed.

2. The expression profiles of Irx genes in A. hentzi (fig. S11-13) are mentioned only in the Discussion section. This data should be stated much earlier in the Results section. Similar profiles for the 16 priority genes are informative and should be presented if possible.

3. The authors concluded that waist-less played a role in dorso-ventral patterning. However, the phenotypes of the waist-less knockdown embryos were analyzed only in the late stages when morphological abnormalities were noticeable. It is possible that the described phenotypes are secondary effects following the defects in germ band morphology. Furthermore, the waist-less gene is expressed in the embryo at late stage 5, as shown in Sci. Adv. 2020; 6, eaba7261 (Data file S5-7, aug3.g8295). The position of early expression along the AP axis seems to be near the boundary between the prosoma and opisthosoma. Knockdown embryos should be examined for the earliest defects; re-examining gap phenotypes and dorso-ventral patterning defects in earlier embryos would be interesting. Additionally, how is the AP and DV pattern in the embryo shown in Fig. S4?

4. Although pannier2 gene was included in the list of 16 priority genes for functional screening (Table S1), it was presented only in the last part without mentioning the screening. Consistent statements are required to address this point.

5. Very beautiful pictures showing the expression of Irx genes of P. opi and P. tep were mentioned only in the Discussion section. The data should be stated in the Results section.

Minor points:

1. The reasons for the selection of the 16 genes (and unselection of other genes) for functional screening (p5, l17-120) should be shown concretely, for example using tables that display quantified expression levels in each sample.

2. The criteria for assessing phenotypes during the screening should be included (p6, 138). Were the embryos stained for the expression of certain genes or examined morphologically? At which stages were the embryos examined?

3. As in p6, 144, the names of genes have been confused. It would be helpful to attach LOC numbers in the NCBI database for listed genes.

4. The name Irx4 may not be correct (p8, 185). Verify and revise as necessary.

5. I understand that the staining shown in Fig. 3 and 4 was performed using mixed probes. It is better to include a note on this and briefly explain the expression patterns of each gene.

6. Fig. 5E (p13, 313) and Fig. 5F. 5G (p13, 315) need to be rechecked and revise for accuracy.

7. Regarding the phenotype of pnr2 knockdown embryo, the graph (Fig. 6E) showed that only a small percentage of embryos exhibited this phenotype. How and when were the embryos counted? Did embryos without this phenotype exhibit decreased pnr2 expression?

8. I cannot understand Fig. 7. Do the illustrations on the right and left show differences in the developmental stages? This legend should be rewritten with more detail.

9. How many times was the dsRNA injection performed (p23, p24)?

Rev. 2: Ab. Matteen Rafiqi - note that this reviewer has signed his review

The paper "A taxon-restricted duplicate of Iroquois3 is required for patterning the spider waist." Written by Setton and co-workers present data on the differences in transcription between the prosoma and the opisthosoma using transcriptomic analysis. They further identify 16 candidates from the transcriptome data that appear promising in defining the PO axis and the PO boundary. They conduct RNAi and HCR-based in situ hybridization experiments and show that irx4 is involved in the regulation of the dorsoventral axis in the precise location of the PO boundary, which results in a waist-less loss of function phenotype. Furthermore, the analysis of markers post RNAi shows that the components of the DV axis regulation such as sog and pnr homologs are affected in the region.

The paper is well-written and presents interesting data on the development of spiders. This is promising given the technical challenges in this type of organism, where experimental tools are not as well developed as in model organisms.

In my opinion, this paper is fit for publication with minor corrections in the text.

Minor revisions needed:

One of the gene names is not consistent and is interchangeably swapped between irx4, ptep-irx4, and waist-less. The authors should clarify this early on in the text and use the same name in the main text as well as supplementary tables.

lines 157, 159, and 177: The usage of the words 'exemplars' is not conventional and could confuse a wider audience.

lines 87, 114: Like my previous comment, the usage of the word 'triangulate' appears wrong. The term could be replaced by a more widely used term.

Lines 313, and 315: The reference to the figure is wrong, there are only As and Bs in Figure 5.

Line 578: The sentence should read, "The negative controls were injected…."

Lines 318-320: Could be revised to something similar in lines 387-388. The term territory-specific makes the phenotype more understandable.

---

## [Decision Letter · Decision Letter 2]

21 Jun 2024

Dear Dr Setton,

Thank you for your patience while we considered your revised manuscript entitled "A taxon-restricted duplicate of Iroquois3 is required for patterning the spider waist" for publication as a Research Article at PLOS Biology. This revised version of your manuscript has been evaluated by the PLOS Biology editors, the Academic Editor and one of the original reviewers.

Based on the reviews, we are likely to accept this manuscript for publication, provided you satisfactorily address the data and other policy-related requests stated below.

We expect to receive your revised manuscript within two weeks. 

*Published Peer Review History*

*Press*

Sincerely,

Ines

--

Ines Alvarez-Garcia, PhD

Senior Editor

PLOS Biology

Fig. 2J; Fig. 7E; Fig. S2; Fig. S3; Fig. S11A, B; Fig. S12A, B and Fig. S13A, B

***Please also indicate in the relevant figure legends where the data from the transcriptional profiles can be found and make sure the data you have deposited is made publicly available at this stage.

CODE POLICY

Reviewers' comments:

Rev. 1:

The authors have conscientiously addressed most of the concerns I raised, and the manuscript has been satisfactorily revised. Consequently, this study now meets the standard of quality necessary for publication in the journal.

---

## [Editor Report · Decision Letter 3]

26 Jul 2024

Dear Dr Setton,

Thank you for the submission of your revised Research Article entitled "A taxon-restricted duplicate of Iroquois3 is required for patterning the spider waist" for publication in PLOS Biology. On behalf of my colleagues and the Academic Editor, Yi-Hsien Su, I am delighted to let you know that we can in principle accept your manuscript for publication, provided you address any remaining formatting and reporting issues. These will be detailed in an email you should receive within 2-3 business days from our colleagues in the journal operations team; no action is required from you until then. Please note that we will not be able to formally accept your manuscript and schedule it for publication until you have completed any requested changes.

PRESS

Sincerely, 

Ines

--

Ines Alvarez-Garcia, PhD

Senior Editor

PLOS Biology
